# Trade and Embodied CO_2_ Emissions: Analysis from a Global Input–Output Perspective

**DOI:** 10.3390/ijerph192114605

**Published:** 2022-11-07

**Authors:** Xinsheng Zhou, Qinyang Guo, Yuanyuan Wang, Guofeng Wang

**Affiliations:** 1Faculty of International Trade, Shanxi University of Finance and Economics, Taiyuan 030006, China; 2School of Public Administration, Shanxi University of Finance and Economics, Taiyuan 030006, China

**Keywords:** trade embodied carbon, input–output, continent, economic cooperation organization, country

## Abstract

Global trade drives the world’s economic development, while a large amount of embodied carbon is transferred among different countries and regions. Based on a multi-regional input–output model, the trade embodied carbon transfers of bilateral trade between 185 countries/regions around the world were calculated. On the basis, regional trade embodied carbon transfer patterns and major national trade patterns in six continents, eight major economic cooperation organizations, and six representative countries/regions were further analyzed. The results showed that Europe was the continent with the largest embodied carbon inflows from trade and Africa was the continent with the largest embodied carbon outflows from trade. China was the country which had the largest embodied carbon outflows from trade, while the United States, France, Japan, and Germany were countries which had embodied carbon inflows from trade. OECD, EU, and NAFTA were the economic cooperation organizations with embodied carbon inflows from trade, while BRICS, SCO, RCEP, OPEC, and ASEAN were economic cooperation organizations with embodied carbon outflows from trade. Developed countries such as the United States, France, and the United Kingdom protected their environment by exporting high-value products and importing low-value and carbon-intensive products. Developing countries such as China and Russia earned foreign exchange by exporting carbon-intensive and commodity products at a huge environmental cost. In contrast, Germany, China, and Russia played different roles in the global industrial chain, while Germany exchanged more trade surpluses at lower environmental costs. Therefore, for different countries and regions, their own industries should be actively upgraded to adjust the import and export structure, the cooperation and coordination in all regions of the world should be strengthened, and the transfers of embodied carbon needs to be reduced to make the trade model sustainable.

## 1. Introduction

Climate change has become an indisputable fact. On 28 February 2020, the Intergovernmental Panel on Climate Change (IPCC) released the Sixth Assessment Report (AR6) Working Group II Report, Climate Change 2022: Impacts, Adaptation and Vulnerability. The report noted that about 50–75% of the population in the world may be exposed to life-threatening climatic conditions by 2100 due to extreme heat and humidity. Climate change would exert increasing pressure on food production and access, and undermine food security and nutrition, especially in vulnerable areas. The temperature rise must be controlled below 1.5 °C, and once this critical line is exceeded, the earth will suffer irreversible damage, the risk of extinction of most organisms will be greatly increased, and the frequency of droughts, floods, and tsunamis will be more frequent. In order to refrain from these risks, global carbon emissions should be reduced by 45% by 2030 and zero emissions need to be achieved by 2050. Although it is a pessimistic assessment of the present and future impacts, if actions can be taken in time, there is still time to avoid the worst of the disaster [1]. According to the WTO database, the world trade imports rose from USD 6647.5 billion to USD 22,518.8 billion in 2000–2021, with an increase of about 2.39 times. The world trade exports increased from USD 6454 billion to USD 22,283.8 billion, with an increase of about 2.45 times [2]. With the rapid growth of trade, carbon emissions have also shifted a lot. Therefore, the study of embodied carbon transfers in global trade is a focus of the goal of controlling rising global temperature.

Scholars from China and other countries have studied the issue of embodied carbon transfers from the relationship of embodied carbon transfers among countries [3], regions, and the world [4], the measurement database, and the measurement method [5]. From the perspective of embodied carbon transfers among countries around the world, scholars paid attention to the embodied carbon transfers in trade in major economies [6]. For example, China, Japan, and South Korea were closely linked in the global industrial chain. Although China had maintained a trade deficit with Japan and South Korea, the carbon intensity of its export products was higher, and it was the country with net outflows of embodied carbon [7]. Germany had a surplus in trade with the United States, and was in the position of net outflows of embodied carbon in terms of trade with the United States. The CO_2_ emissions of the two countries had a tendency to shift to developing countries [8]. There had been a long-standing trade deficit between China and the United States [9]; China was in the position of embodied carbon outflows in trade at a huge environmental cost [10]. Carbon dioxide emissions from China–Brazil trade had maintained a steady growth, and China was a net carbon dioxide emitter [11]. European economies with lower development levels (many Eastern and Southern countries) seemed to be still on the rising trend of carbon emission, while the more developed ones seemed to be on the decreasing trend of carbon emission between 2005 and 2015 [12]. Chinese scholars constructed a spatial correlation network of net carbon transfer in global trade including 185 countries (regions) in 2000 and 2015 and used multi-dimensional network measurement indicators to comprehensively examine the characteristics of change of the global trade embodied carbon emission network and the role and functional characteristics of the network sectors [13,14]. In 2014, British scholars conducted research about trades and embodied carbon emissions between 195 countries for 1080 products in 2006; the results highlighted that embodied carbon was focused in regional trade [15]. At the same time, the emissions embodied in world trade in 2004 were estimated to be between 4.4 Gt and 6.2 Gt of CO_2_ [16].

From the perspective of estimated data sources, it mainly included China Statistical Yearbook, Eora [17], the WIOD, EXIOBASE, OECD, and ICIO databases, etc. From the perspective of measurement methods, some scholars used the multi-regional input–output model [18] to measure the embodied carbon in trade between South Korea and China. It was found that the carbon emission coefficient of South Korea had been lower than that of China, but the carbon emission coefficient of China had declined faster than South Korea [19]. Some scholars had also calculated the embodied carbon in multi-country trade based on the hypothesis extraction method [20]. In addition, some scholars had calculated the embodied carbon in China’s import and export trade through the value-added trade method [21]. The multi-regional input–output (MRIO) model considered the technological and economic structure of different countries, comprehensively described the complete production chain among various sectors of each country, and distinguished the different directions of imported products. Therefore, the model became the most effective method to measure the multinational and global carbon emissions. The MRIO model still faced great challenges in data acquisition, model construction, and department aggregation. Data sources and compilation methods were different, especially in the calculation of carbon emission factors; there were obvious differences between different databases, which reduced the comparability and credibility of accounting results to a certain extent.

Based on the existing literature, there was still a lack of analysis of the transfer characteristics of the embodied carbon in global trade. The marginal contribution of this paper was as follows: Firstly, the embodied carbon transfers in trade among 185 countries/regions in the world were calculated by a multi-regional input–output model, and the transfer situation of embodied carbon in global trade was analyzed. Secondly, the embodied carbon transfers in trade of six continents were analyzed, and the division of labor of each continent in the global trade chain was observed. Thirdly, the embodied carbon transfers in trade of major economic cooperation organizations were analyzed. Fourth, the partners of major embodied carbon transfers of six developed and developing economies were detected. Fifth, the patterns of embodied carbon transfers and trade in goods among major economies were analyzed.

## 2. Methods and Data

### 2.1. Methods

The bilateral input–output model is a tool for measuring the embodied carbon in trade, and it has been widely used in the research related to interregional carbon transfers [22,23]. With the consideration of the technological gaps and intermediate products in various countries, a multi-regional input–output model (MRIO) is used [24] in this study to calculate the embodied carbon transfers in international trade of 185 countries [25]. The core of the MRIO is the input–output table (Table 1). From the vertical analysis of the input–output table, the intermediate input part of the input–output table represents that the output of a certain sector in a certain country/region is the intermediate input of a certain sector in other countries/regions; the output of a certain sector in a certain country/region in a final demand is the final demand of a certain sector in a certain country/region [26]. From the horizontal analysis of the input–output table [27], the input of a certain sector in an intermediate input country/region is the output of a certain sector in a certain country/region [28].

It is supposed that there are a total of R countries/regions in the world, each country/region can be divided into N production sectors, and the sum of all industrial sectors in each country/region is the world input–output table for R countries. $Z^{EF}$ is the R × R dimensional intermediate input matrix (E, F = 1, 2, ....., R) of all industrial sectors in country E to all industrial sectors in country F, where the matrix element $z_{I}{}^{EF}$ (I = 1,2, ..., N) refers to the intermediate input of a sector in country E to a sector in country F. $Y^{EF}\;$ is the N × M dimensional final demand matrix of country E to country F, where the matrix element $y_{L}^{EF}$ (L = 1, 2, ..., M) refers to the final use of a sector in country E for the final demand of M sectors in country F. $X^{E}\;$ represents the total output of country E, where the matrix element $x_{F}^{E}$ refers to the output vector of the R × 1 dimension. By adding carbon emissions data to the table, a 1 × NR dimensional emission matrix of CO_2_ emissions of N sectors in country R is obtained. $Q_{I}{}^{E}$ (I = 1, 2, ..., N) is obtained, where the matrix element $q_{l}{}^{E}$ (E = 1, 2, ..., R) refers to the CO_2_ emissions of a certain sector in the country E.
(1)$$Z^{EF}=\left[\begin{array}{ccc} z^{11} & \cdots & z^{1R}\\ \vdots & \ddots & \vdots \\ z^{R1} & \cdots & z^{RR} \end{array}\right],\;Y^{EF}=\left[\begin{array}{ccc} y^{11} & \cdots & y^{1R}\\ \vdots & \ddots & \vdots \\ y^{R1} & \cdots & y^{RR} \end{array}\right]$$
(2)$$X^{E}=\left[\begin{array}{c} x_{}^{1}\\ \vdots \\ x_{}^{NR} \end{array}\right],\;Q^{E}=\left[q^{1}q^{2}\ldots ..q^{R}\right]$$

The general equilibrium formula of the input–output model is that the total input is equal to the total output.
(3)$$\left\{ \begin{array}{l} Z_{11}{}^{11}+Z_{12}{}^{11}+\ldots ..Z_{1N}{}^{1R}+Y_{11}{}^{11}\ldots ..Y_{1N}{}^{1R}=X^{1}\\ \ldots \\ Z_{N1}{}^{11}+Z_{N2}{}^{11}+\ldots ..Z_{NN}{}^{1R}+Y_{N1}{}^{11}\ldots ..Y_{NN}{}^{1R}=X^{N}\\ Z_{11}{}^{21}+Z_{12}{}^{21}+\ldots ..Z_{1N}{}^{2R}+Y_{11}{}^{21}\ldots ..Y_{1N}{}^{2R}=X^{N+1}\\ \ldots \\ Z_{N1}{}^{R1}+Z_{N2}{}^{R1}+\ldots ..Z_{NN}{}^{RR}+Y_{N1}{}^{R1}\ldots ..Y_{NN}{}^{RR}=X^{NR} \end{array}\right.$$

Based on the volume of carbon emissions and the total output of each sector in each country, the ratio of the above two is the carbon consumption coefficient $G_{i}{}^{E}$ (E = 1,2, ..., R, I = 1, 2, .... N) for each sector in each country, and the carbon coefficient matrix can be obtained as follows.
(4)$$G^{E}=\left[g^{1}g^{2}\ldots .g^{R}\right],\;g^{E}=\left[g_{1}^{E}g_{2}^{E}\ldots .g_{N}^{E}\right]$$

The direct consumption coefficient A is the input of each sector in each country to a sector in another country divided by the total output, and a 1 × NR dimensional matrix $A_{l}{}^{E}$ (E = 1, 2, ..., R, l = 1, 2, ..., N) can be obtained as follows.
(5)$$A^{E}=[a^{1}a^{2}\ldots .a^{R}],\;a_{i}^{E}=[a_{1}^{E}a_{2}^{E}\ldots .a_{N}^{E}]$$

The Leontief inverse matrix ${\left(I-A\right)}^{-1}$ is calculated by the direct consumption coefficient. The transfer value of embodied carbon in R countries and N sectors is the multiple of the direct carbon consumption coefficient, the Leontief inverse matrix, and the export value. $C^{EF}\;$ is the embodied carbon transfer of export trade from country E to country F, which is a NR × NR dimensional matrix with a diagonal of 0. By summing up all industrial sectors in each country, the R × R dimensional matrix of total embodied carbon transfers of export trade from country E to country F ($C^{EF}$) is obtained as follows.
(6)$$C^{EF}={\left(I-A^{EE}\right)}^{-1}[A^{EF}X^{F}+Y^{EF}],\;C^{EF}=\left[\begin{array}{ccc} C^{11} & \cdots & C^{1R}\\ \vdots & \ddots & \vdots \\ C^{R1} & \cdots & C^{RR} \end{array}\right]$$

According to the transfer value of embodied carbon between countries, the net transfer value of embodied carbon from country E to country F ($\Delta C^{EF}$) can be calculated, which is a R × R dimensional symmetry matrix with a diagonal of 0. $\Delta C^{EF}$ > 0 means country E is an embodied carbon outflow country and bears part of the carbon emissions for Country F.
(7)$$\Delta C^{EF}=C^{EF}-C^{FE}$$

The embodied carbon outflow country bears more carbon emission hazards for the embodied carbon inflow country. Therefore, the embodied carbon outflow country is regarded as the “Pollution Shelter” for the embodied carbon inflow country.

### 2.2. Data

The World Input-Output Eora Database [29], which provides a high-resolution time-series input–output table containing environmental and social satellite data for 190 countries and sectoral transfers among 15,909 sectors [30,31], was used in this study. The compressed data package in 2016 was downloaded on the Eora website, and according to the Eora 26 Structure file, 190 world input-output databases in 2016 were composed of multiple data files [32]. In this study, the data of Chinese mainland, Taiwan, Hong Kong, and Macau were combined into one country and the data of former Soviet Union was removed [33]. Then, a 4997 × 5924-dimensional matrix of 185 countries/regions for 26 sectors was obtained, and a 1 × 4810-dimensional matrix of the direct carbon emission data from each sector of each country from the environmental satellite account was extracted [34]. Moreover, 185 countries were divided into six continents to calculate embodied carbon transfers within and between continents, and 8 major ECOs were extracted to calculate embodied carbon transfers between OECD and continents [35]. The net transfers of embodied carbon in 185 countries were ranked, and the transfer structure and major trading partners of 6 typical net carbon transfer countries were analyzed. The GDP data of countries around the world in 2016 were collected from the official website of the World Bank, and the GDP of each country was mapped to study the embodied carbon transfers from trade of the 8 major economic cooperation organizations (Table 2). The export and import data of major economies in goods trade in 2016 was collected from the UN Comtrade database to calculate the balance of commodity trade of each country and analyze the trade patterns of major economies.

## 3. Results

### 3.1. Embodied Carbon Net Transfers from Intercontinental Trade

In 2016, total global imports and total global exports reached USD 15.8 trillion and USD 15.5 trillion, respectively. Meanwhile, total embodied carbon emissions in trade were 57,749.27 Mt. The carbon emission amount adopted in this paper included short-cycle CO_2_ generated by biomass combustion, which was usually carbon dioxide emissions generated by land use, land use change, and fire. Therefore, it was higher than the carbon emission data published by the International Energy Agency, and the calculation result of carbon embodied in trade was higher. From the perspective of six major continents, Asia was the continent with the largest embodied carbon emissions, while Oceania had the smallest embodied carbon emissions. Figure 1 shows the embodied carbon transfers within and among continents.

The embodied carbon emissions of the continents from largest to smallest were Asia (52.88%), Europe (22.69%), South America (19.71%), Africa (14.02%), North America (13.08%), and Oceania (1.60%). Asia was most involved in globalization and trades closely with all regions of the world, thus generating the most embodied carbon emissions. With more countries and closer trade, Asia had the most embodied carbon transfers in trade with countries within the continent. Meanwhile, Oceania contained fewer countries and thus had the least embodied carbon transfers with countries within the continent. The embodied carbon transfers in trade within the continents from largest to smallest were Asia (27,547.78 Mt), Europe (10,413.59 Mt), Africa (6619.38 Mt), North America (6130.60 Mt), South America (1767.70 Mt), and Oceania (651.09 Mt). Asia and Africa transferred a large amount of embodied carbon to other developed regions through trade. Europe had more developed countries, while there were more developing and underdeveloped countries in Asia and Africa. Generally, the proportion of carbon-intensive industries in developed countries was lower, while developing countries were opposite. The top five embodied carbon transfers from continent-to-continent trade were Asia to Europe (982.56 Mt), Africa to Europe (655.16 Mt), Europe to Asia (458.81 Mt), Africa to Asia (454.26 Mt), and Asia to North America (437.11 Mt).

Overall, Europe, North America, Oceania, and South America were in the carbon-benefiting position in global trade, while Africa and Asia bore a lot of carbon reduction pressure due to more carbon emissions in global trade and the responsibility of export of low-end carbon-intensive products. Africa was the continent with the most net outflows of embodied carbon from trade in the world, with an outflow volume of 1149.79 Mt. Europe was the continent with the most net inflows of embodied carbon in trade in the world, with an inflow volume of 1141.27 Mt. It was shown that Africa bore a large amount of carbon emissions in global trade, while Europe transferred out a large amount of carbon emissions in global trade. The similarity between Africa and Europe was that the gaps of embodied carbon transfers between import and export were relatively large. However, Asia was different from Africa and Europe. The import and export of Asia involved a large amount of embodied carbon transfers, and Asia had the largest embodied carbon transfers in import and export trade among the six major continents, but the gaps were relatively small, resulting the small net transfer. Africa transferred nearly 84.45% of the embodied carbon from trade to Europe and Asia.

Africa exported many carbon-intensive products and imported high-value low-carbon intensive products, corresponding to the poor development of economy and science and technology in Africa and the advanced development of economy and science and technology in Europe. Asia mainly transferred out embodied carbon to Europe and North America, and received carbon inflows from Africa. North America mainly transferred out a small amount of embodied carbon from trade to Europe, and received a certain amount of embodied carbon inflows from other continents. Europe mainly received embodied carbon inflows from four continents such as Africa and Asia, and transferred out a small amount of embodied carbon in trade to Oceania. South America received embodied carbon inflows from Africa and Asia, and transferred a certain amount of embodied carbon from trade to other continents.

### 3.2. Embodied Carbon Transfers of Eight Major Global Economic Cooperation Organizations

From the perspective of major economic cooperation organizations, which mainly included ASEAN, NAFTA, SCO, OECD, EU, RCEP, OPEC, and BRICS, embodied carbon transfers in trade were further studied. Economic cooperation organizations had more concentrated trade and embodied carbon transfers in trade of different economic cooperation organizations were more representative. For example, the OECD was composed of market economy countries, and the BRICS countries represented the world’s emerging markets. As is shown in Figure 2, economic developments in various regions of the world and embodied carbon transfers in major economic cooperation organizations could be observed. In 2016, the top ten countries and regions in terms of GDP were the United States, China, Japan, Germany, the United Kingdom, France, India, Italy, Brazil, and Canada. Countries such as China and India were net outflow countries of embodied carbon in trade, bearing carbon emission pressures for other countries. On the contrary, the United States, Japan, and other countries were net inflow countries of embodied carbon in trade.

OECD, EU, and NAFTA were economic cooperation organizations with net inflows of embodied carbon in trade. The members of these organizations were mainly responsible for the export of low-carbon intensive products upstream of the global industrial chain, transferring the manufacture of many high-carbon intensive products to other regions and importing from these regions. OECD, whose members were mainly in North America and Europe, had major market economy countries. The top ten countries in terms of GDP, in addition to China, India, and Brazil, were in the organization. OECD was a typical net inflow economic cooperation organization of embodied carbon in trade. This was mainly because trade among market economy countries involved a large amount of embodied carbon emissions. Embodied carbon emissions among its internal member countries were 16,723.5 Mt and it received net inflows of 2053.61 Mt of embodied carbon in trade. The net transfers of embodied carbon from trade among OECD and major continents from largest to smallest were Asia (−1098.50 Mt), Africa (−720.05 Mt), Europe (−161.51 Mt), South America (−42.77 Mt), North America (−27.43 Mt), and Oceania (−3.35 Mt). Asia and Africa transferred out a large amount of embodied carbon to OECD, while OECD transferred out the carbon emission pressures to Asia and Africa.

EU mainly occupied the upstream high-end industry in the global industrial chain and exported high-tech and low-carbon intensive green products. The embodied carbon emissions within the organization were 6019.56 Mt, and it received embodied carbon inflows of 914.23 Mt from countries around the world. Most of the EU members were the earliest countries to achieve industrialization, and were responsible for a large part of the global accumulated carbon emissions. Therefore, EU should play a leading role with an industrialized power to actively lead Africa, Asia, and others to an industrial upgrading energy transformation. EU imported a large number of carbon-intensive products from Africa, while also maintaining frequent import and export trade with Asia. The net transfers of embodied carbon in trade among EU and major continents from largest to smallest were Africa (−475.41 Mt), Asia (−330.29 Mt), Europe (−71.58 Mt), North America (−18.43 Mt), South America (−16.99 Mt), and Oceania (−1. 53 Mt). The three members of NAFTA were three representative powers of North America and the trade relations among them.

NAFTA was a net inflow economic cooperation organization of embodied carbon in trade; a large amount of carbon emissions was transferred out to the rest of the world from trade, especially to many developing countries and backward countries. The embodied carbon emissions within the organization were 5768.60 Mt, and it received embodied carbon inflows of 256.88 Mt from other countries around the world. Therefore, the carbon emissions of product manufacture were outside the region, while the products were consumed within the region. The net transfers of embodied carbon in trade among EU and major continents from largest to smallest were Asia (−142.37 Mt), Africa (−76.21 Mt), South America (−18.64 Mt), North America (−18.64 Mt), Oceania (−15.23 Mt), and Europe (14.21 Mt). It could be seen that NAFTA transferred out a small amount of embodied carbon to Europe, while it received embodied carbon net inflows from other continents.

OPEC, ASEAN, RCEP, BRICS, and SCO were economic cooperation organizations with net outflows of embodied carbon in trade. A large number of carbon-intensive products were produced and exported from members in these organizations. The industrial development of members in OPEC, which mainly earned foreign exchange by selling carbon-intensive products such as commodities, were relatively poor. As the major oil export organization, the original intention of the establishment of OPEC was to protect the economic interests of the major oil-producing countries in the third world. OPEC transferred out 178.62 Mt embodied carbon to the world, bearing a certain amount of carbon emission reduction pressures for the world. OPEC mainly transferred embodied carbon to Europe, North America, South America, and Oceania, and received embodied carbon inflows from Asia and Africa. The net transfers of embodied carbon in trade among OPEC and major continents from largest to smallest were Europe (192.94 Mt), Asia (−60.31 Mt), North America (39.05 Mt), Africa (−16.54 Mt), South America (16.28 Mt), and Oceania (7.20 Mt).

With rapid development in recent years, ASEAN was the third largest economy in Asia and mainly transferred out 308.61 Mt embodied carbon to the world. ASEAN had more than 600 million population, low labor costs, abundant resources, and huge potential in the consumer market, which had helped it become one of the largest markets in the world economy. The net transfers of embodied carbon in trade among ASEAN and major continents from largest to smallest were Europe (178.17 Mt), Asia (148.23 Mt), Africa (−108.52 Mt), North America (64.90 Mt), Oceania (19.38 Mt), and South America (6.45 Mt). It could be concluded that only Africa transferred out a certain amount of embodied carbon to ASEAN, sharing some carbon reduction pressures for ASEAN. The members of BRICS, which accounted for nearly a quarter of the world’s GDP and enjoyed a rapid economic growth rate, included Brazil, Russia, China, South Africa, and India. These countries were in the middle and lower reaches of the global industrial chain and were experiencing ongoing gradual industrial transformation. Their production and export of commodities were carbon-intensive products, which had a high degree of pollution.

In 2016, BRICS transferred out 623.06 Mt embodied carbon to the world. The net transfers of embodied carbon in trade among BRICS and major continents from largest to smallest were Europe (386.78 Mt), Asia (289.69 Mt), Africa (−227.50 Mt), North America (99.40 Mt), Oceania (55.22 Mt), and South America (19.47 Mt). It can be found that the members of BRICS mainly transferred out embodied carbon to Europe, Asia, North America, Oceania, and South America, bearing a certain amount of carbon emissions for these regions, while Africa bore part of the carbon emissions for the BRICS countries.

With the largest economic and trade scale, RCEP was a free trade zone that occupied about 30% of the global population and about 30% of the global GDP. With a large amount of global carbon emissions generated, the members of RCEP transferred out 206.17 Mt embodied carbon to the world. The net transfers of embodied carbon in trade among RCEP and major continents from largest to smallest were Europe (368.28 Mt), Africa (−281.40 Mt), North America (128.72 Mt), Asia (−29.24 Mt), South America (23.65 Mt), and Oceania (−3.85 Mt). RCEP produced a large number of carbon-intensive products for Europe, North America, and South America, and imported a large number of carbon-intensive products from Africa.

The total area of SCO members accounted for 60% of the Eurasian continent, the total population was 40% of the world’s population, and the GDP occupied more than 20% of the world’s GDP. SCO, whose members were in the middle and lower reaches of the global industrial chain, was a high pollution and high carbon emission concentration area in the world. In 2016, it transferred out 682.36 Mt embodied carbon to the world. The net transfers of embodied carbon in trade among SCO and major continents from largest to smallest were Europe (402.41 Mt), Asia (260.38 Mt), Africa (−1 55.36 Mt), North America (93.57 Mt), Oceania (54.15 Mt), and South America (2721 Mt).

### 3.3. Embodied Carbon Transfers of Major Economies

From the perspective of countries to observe the pattern of global embodied carbon transfers, it could be found that the net transfers of embodied carbon were highly concentrated in several major economies, such as China, the United States, Japan, France, Germany, and Russia. More attention should be paid to the embodied carbon transfers among these major countries, and measures should be taken to promote the reduction of carbon emissions. Figure 3 shows the top 65% of trading partners for net carbon transfers of six major economies.

As the “world’s factory”, China, whose net embodied carbon transfers accounted for 10.17% of the total net transfers in the world, was the country with the largest embodied carbon transfers. Since the accession to the WTO in 2001, China had been integrated into the global industrial chain. Many highly polluting industries and processing trade had been transferred to China. However, with the development of China, the cost of environmental pollution in China had gradually increased, the relevant laws had been continuously improved, and it had become unprofitable to set up factories in China to produce highly polluting products. On the contrary, India and Southeast Asia still had low labor costs and environmental pollution costs, thus an increasing number of factories, especially clothing, textiles, and footwear, had been transferred to India, Cambodia, and other places. China was responsible for the processing and exporting of low and medium carbon-intensive products in the global industrial chain. As the country with the largest carbon emissions in the world, China had taken the initiative to assume the responsibility of a major country and actively proposed the carbon peak and carbon neutrality goals, which were the most challenging carbon reduction tasks in human history. 

The top ten trading partners of embodied carbon net transfers in China from largest to smallest were South Korea (169.89 Mt), Japan (120.32 Mt), the United States (68.57 Mt), Germany (59.59 Mt), India (−40.04 Mt), Singapore (36.34 Mt), Australia (34.79 Mt), France (26.97 Mt), Switzerland (26.53 Mt), the United Kingdom (25.55 Mt), and Cambodia (−23.57 Mt). It could be concluded that China transferred out embodied carbon to other trading partners, bearing many carbon emissions for them, except for India and Cambodia. The main sectors of embodied carbon net outflow in China were electrical and machinery; textiles and wearing apparel; petroleum, chemical, and non-metallic mineral products; metal products; and other manufacturing and other sectors. It was 403.92 Mt, 215.41 Mt, 138.90 Mt, 86.76 Mt, and 85.52 Mt, respectively. In electricity, gas, and water; construction; and agriculture, China’s net trade embodied carbon inflows were −146.22 Mt, −52.04 Mt, and −47.45 Mt, respectively.

In view of the economic development, traditional energy and industrial structures produced a huge waste of China’s economic development, having a risk of falling into a “development trap”. It was worth considering that China had boomed in photovoltaic, 5G, new energy vehicles, and UHV, and these green transformations of related industries meant new business opportunities and new economic growth momentum. However, it should also be realized that China still has a long way to go, as thermal power generation still has a high proportion in China. A bulletin of the National Bureau of Statistics in 2021 stressed that the coal production and consumption in China both had huge problems; the data of coal production and marketing showed the complexity and arduousness of the carbon peak and carbon neutrality goals.

The United States, whose net embodied carbon transfers accounted for 7.96% of the total net transfers in the world, was the country with the largest embodied carbon net inflows. As the largest developed country in the world, the United States was mainly involved in high-end manufacturing, high value-added, and low-carbon intensive products in the global industrial chain. Its production, processing, and manufacturing of high-carbon intensive products had been mostly transferred to other countries, which had greatly reduced the pressure of carbon emission for it. The top ten trading partners of embodied carbon net transfers in the United States from largest to smallest were China (−68.57 Mt), Mexico (−49.11 Mt), India (−24.23 Mt), Guinea (−22.77 Mt), Indonesia (−21.10 Mt), Singapore (−19.13 Mt), Canada (−16.94 Mt), Central Africa (−15.61 Mt), Australia (−14.99 Mt), and Malaysia (−12.64 Mt). It could be concluded that the United States transferred more carbon emissions pressures to Mexico, Canada, Asia, and Africa, and imported carbon-intensive products from these countries, thus protecting its environment. The main sectors of embodied carbon net inflow in the United States were petroleum, chemical, and non-metallic mineral products; textiles and wearing apparel; electricity, gas, and water; other manufacturing; and food and beverages and other sectors, which were −111.03 Mt, −76.60 Mt, −69.35 Mt, −40.26 Mt, and −25.22 Mt, respectively.

For developing countries, they had borne more carbon pressures for the United States, causing their own environmental destruction. However, it also helped them to integrate the global industrial chain, maintain a rapid growth of the domestic economy, and accumulate more foreign exchange and funds for the industrialization transformation. As the leader of the global industrial chain, the United States was supposed to play an important role in global climate governance, lead developed countries to actively increase energy transition and climate adaptation to green industries and developing countries, give full play to the advantages of being a financial power, and ensure sufficient investment and financing for green industries and energy transformation.

France was the country with the second largest embodied carbon net inflows. Europe had begun the process of energy transition and carbon reduction since the London smog in 1952. Nowadays, European countries had occupied the export of upstream products in the global industrial chain and developed their economies at a lower cost of environmental damage. The top ten trading partners of embodied carbon net transfers in France from largest to smallest were Central Africa (−51.38 Mt), Guinea (−25.55 Mt), China (−22.22 Mt), Cameroon (−11.19 Mt), Chad (−10.00 Mt), the Democratic Republic of the Congo (−6.47 Mt), India (−5.04 Mt), Ivory Coast (−4.27 Mt), Germany (−3.43 Mt), and Mali (−3.33 Mt). 

It was interesting that France mainly received embodied carbon inflows from African countries. This was because France had long been the most influential country in Africa, the mining energy of Africa had long been controlled by large French multinational companies, and the African franc was one of the major currencies in circulation. Western Europe had always been relatively scarce in fossil fuels; even in the famous Ruhr region, the coal mines had a higher proportion of poorer-quality lignite coal, and the North Sea oil fields were facing depletion. As a big nuclear power country and the most radical leader of the energy revolution in Western Europe, France was close to phasing out coal power. The main sectors of embodied carbon net inflow in France were mainly in petroleum, chemical, and non-metallic mineral products; agriculture; electricity, gas, and water; re-export and re-import; and food and beverages and other sectors, which were −111.03 Mt, −76.60 Mt, −69.35 Mt, −40.26 Mt, and −25.22 Mt, respectively.

Japan was the country with the third largest embodied carbon net inflows. China had been the largest trading partner of Japan since 2004. For a long time, China had maintained a high import demand for mechanical and electrical products and automobile products of Japan, which reflected great advantages of Japan in high-tech products as a traditional scientific and technological power. Meanwhile, garment products and low-end mechanical and electrical products of China had been exported to Japan for a long time, which made China and Japan have huge embodied carbon transfers from trade. In addition, Japan had also built a triangular trade structure in the East Asian region centered on China through investment. It was achieved by first establishing intermediate product and material bases in the Four Tigers region, then investing in the establishment of processing and assembly factories in China and the ASEAN region, and finally exporting final products to European countries and the United States. In 2016, Japan received 217.13 Mt embodied carbon inflows. 

The top 65% of the trading partners in terms of net transfers of embodied carbon was concentrated in China, Vietnam, Central Africa, Indonesia, and the United Arab Emirates. It was worth considering that China accounted for 38.77% of the net carbon transfers of Japan. China was the first major partner in terms of the embodied carbon transfers of Japan, which had produced many carbon-intensive products for Japan and had reduced huge pressures for Japan to reduce carbon emissions. Japan’s net embodied carbon outflow was 40.82 Mt in transport equipment, 37.45 Mt in electrical and machinery, and 20.33 Mt in wholesale trade, respectively. Gas and water; food and beverages; petroleum, chemical, and non-metallic mineral products; hotels and restaurants; and textiles and wearing apparel and other sectors were enjoying a net inflow of carbon embodied in trade. They were −222.89 Mt, −116.74 Mt, −76.87 Mt, −63.92 Mt, and −59.29 Mt, respectively.

Germany was the country with the fourth largest embodied carbon net inflows. Over the past few years, Germany had rapidly improved wind power and photovoltaic technology, and had reduced carbon emission pressures. In Germany, renewable energy accounted for more than 50% of net electricity generation, wind power had become the most important source of electricity, and photovoltaic power had surpassed hard coal power. The top ten trading partners of embodied carbon net transfers in Germany from largest to smallest were China (−59.59 Mt), India (−20.66 Mt), Russia (−16.21 Mt), Ukraine (−13.01 Mt), Switzerland (12.15 Mt), South Africa (−8.00 Mt), the Democratic Republic of the Congo (−7.93 Mt), Thailand (−7.06 Mt), Indonesia (6.27 Mt), and Ireland (5.85 Mt).

It could be found that China, India, Russia, Ukraine, South Africa, the Democratic Republic of the Congo, and Thailand transferred out embodied carbon to Germany, bearing carbon emissions pressures for Germany, while Germany transferred out embodied carbon to Switzerland, France, and Ireland, bearing the pressures of carbon emissions for these countries. As early as the Fukushima nuclear accident in 2011, Germany decided to shut down all nuclear power plants by 2022 and accelerated the phase-out of coal power. These measures were aimed at increasing the supply of electricity from renewable sources in Germany from 65% to 80% by 2040. As one of the earliest industrialized countries, Germany had accumulated many carbon emissions, and it was supposed for Germany to share more carbon reduction pressures for the world and serve as a leader in green industries and energy transformation. Germany was mainly in the electrical and machinery and other sectors in the net outflow of embodied carbon, and the transfer value was 9.28 Mt. Re-export and re-import; food and beverages; petroleum, chemical, and non-metallic mineral products; agriculture; and transport and other sectors enjoyed a net inflow of trade embodied carbon. They were −147.75 Mt, −58.11 Mt, −47.93 Mt, −44.60 Mt, and −25.78 Mt, respectively.

As a typical large country of embodied carbon transfers, Russia was a net outflow country of embodied carbon. It had close trade relations with Belarus, Kazakhstan, and China. The top ten trading partners of embodied carbon net transfers in Russia from largest to smallest were Belarus (31.64 Mt), China (−19 98 Mt), Kazakhstan (−17.14 Mt), Germany (16.21 Mt), Ukraine (−14.71 Mt), India (−10.65 Mt), the United States (8.48 Mt), Poland (7.88 Mt), Slovakia (−5.48 Mt), and Italy (5.23 Mt). It could be concluded that China, Kazakhstan, Ukraine, India, and Slovakia mainly transferred out embodied carbon to Russia, while Russia transferred out embodied carbon to Belarus, Germany, the United States, Poland, and Italy to bear a certain amount of carbon emissions for these countries. As a traditional oil resource country, Russia should actively cooperate with developed countries and China and the Eurasian Economic Union to actively promote energy transformation and industrial upgrading and jointly contribute to the global carbon emission reduction.

### 3.4. Trade Patterns of Major Global Economies

From the perspective of trade patterns of major economies, there were four main trade patterns of trade deficit and embodied carbon trade deficit, trade surplus and embodied carbon trade surplus, trade deficit and embodied carbon trade surplus, and trade surplus and embodied carbon trade deficit. According to Figure 4, the carbon trade patterns of major economies could be analyzed based on the trade balance of goods and trade embodied carbon transfer of major economies.

Firstly, it could be observed that China and Russia had similarities in their trade patterns. They all utilized their comparative advantages to gain trade surplus from international trade and earn a certain amount of foreign exchange and capital. At the same time, they exported a large number of carbon intensive low value-added products in the world industry chain, and transferred a large amount of embodied carbon to the world. However, this kind of pattern had certain defect; oil and gas prices were volatile and unstable, and the balance of payments was vulnerable to fluctuations in the international market. Especially when oil prices fell sharply, the country’s ability to resist risks would be greatly reduced. Secondly, India and Mexico had similar trade patterns with a certain trade deficit and the export of carbon-intensive products in the global industrial chain. Thirdly, trade patterns of the United States, France, and the United Kingdom were similar. They all had trade deficits but import carbon-intensive products from the world. Fourthly, trade patterns of the Netherlands, Japan, Germany, South Korea, and Australia were similar. They all got a certain trade surplus in international trade and earned a certain surplus by exporting high-end manufacturing and high-tech industrial products, such as high-end agricultural products in The Netherlands and electronic products and automobiles in Japan.

It was interesting that although China and Germany were both trade surplus countries, their embodied carbon transfers were different, showing that they had different roles in global trade. China was currently in the middle of the global value chain, while the United States was at the top of the global innovation chain. It had shifted its manufacturing links to China, South Korea, and Japan through industrial outsourcing. Japan and Germany were at the high end of the global innovation chain, occupying the high value-added link of the global value chain through “production + innovation” [36]. Germany could exchange more trade surpluses at lower environmental pollution costs and reduce environmental pollution by exporting high value-added products and importing low value-added and high carbon emission products. It was worth considering that China and India had borne large carbon emissions for national development in an unsustainable development pattern. Nowadays, smog issues and large carbon emissions made the Chinese government lay more emphasis on green transformation. China had accelerated the upgrading of the global industrial chain from the middle and lower reaches to the upper reaches and had powerfully supported the development of low-carbon technologies and personnel training.

## 4. Conclusions and Suggestions

### 4.1. Conclusions

International trade was recognized as a major cause of environmental problems. In order to cope with climate change, the situation of embodied carbon transfers in trade needed to be clarified, so as to provide targeted recommendations and strategies. In this study, the latest data from the World Input-Output Eora database was used and the embodied carbon transfers in trade among 185 countries in 2016 were calculated. In addition, the patterns of embodied carbon transfers in trade were studied from different perspectives, including continents, economic cooperation organizations, major economies, and trade patterns of major economies, to analyze the characteristics of embodied carbon transfers in trade. The main results were provided as follows.

Firstly, there were significant differences in embodied carbon transfers in trade of each continent. It indicated that each continent played a different role in the global industrial chain. Continents with backward economic development, which were in the disadvantaged position in embodied carbon transfers in trade, were responsible for the production and export of low-end carbon-intensive products in the global industrial chain. On the contrary, continents with good economic development, which were in the advantaged position in embodied carbon transfers in trade, were responsible for the export of large quantities of high-value products in global trade. The export of low-value carbon-intensive products in the global industrial chain were concentrated in Asia and Africa and the import of Africa involved less embodied carbon transfers. Although the import and export of Asia involved large embodied carbon transfer, the gaps of embodied carbon transfers between import and export in Africa were larger. Compared with Africa, the economic development of Asia was relatively better, thus some carbon-intensive products could be imported from Africa. Carbon emissions were closely related to industrial activities and resource consumption, which were important indicators for evaluating regional sustainable development. Therefore, economically backward areas should actively develop their own science and technology and economy to avoid falling into the trap of sustainable development.

Secondly, OECD, NAFTA, and EU, which were the concentration of upstream product exports in the global industrial chain, were the economic cooperation organizations with net inflows of embodied carbon. OPEC, ASEAN, BRICS, RCEP, and SCO were the economic cooperation organizations with net outflows of embodied carbon, producing many carbon-intensive products for the world. Economic cooperation organizations gathered in developed countries so as to use developed technology, complete industrial chains, and mature management models to “outsource” front-end manufacturing industries with high pollution and high energy consumption to developing regions with low production costs and imperfect environmental protection laws and regulations.

Thirdly, differences in factor endowments between countries led to a global division of labor and international trade and embodied carbon transfers in trade were highly concentrated in major economies. For example, China accounted for 10.98% of the global net transfers of embodied carbon in 2016. China, the United States, France, Japan, Germany, and Russia were the major economies in terms of net transfers of embodied carbon. Trade patterns and embodied carbon transfers among major countries reflected the division of labor of the global industrial chain, and the rational division of labor in the industrial chain could greatly promote the growth of the global economy. Therefore, suggestions could be made according to the transfers of embodied carbon in trade. For emerging economies, such as China and India, the essence of the issue of the emission reduction was the issue of the development. The debate over limiting the growth of emerging and developing economies was bigger than the one over climate. The huge exports and carbon emissions of many emerging economies, including China and India, had in fact created a dilemma for the formulation of environmental policies. On the one hand, emerging economies could take full advantages of their comparative advantages to create economic benefits for countries around the world through trade. On the other hand, many emerging economies, such as China and India, had higher carbon emissions intensities, which in turn further boosted global carbon dioxide emissions. Therefore, countries around the world should strive to achieve green and sustainable development, pay attention to their own development models, avoid falling into the development trap of high environmental damage costs, and gradually reduce the environmental costs of economic development. Moreover, developed countries need to assume more responsibilities for emission reductions.

This paper focused on revealing the flow of carbon embodied in trade between countries, but the results lacked evolution of trade embodied CO_2_ emissions and key analyses for characteristics such as impact effects, causes, and paths. Further research and work were needed to follow up. Research about regional trade embodied carbon transfer patterns and major national trade patterns in continents, major economic cooperation organizations, and representative countries/regions could well reveal the carbon flow embodied in international and regional trade.

### 4.2. Suggestions

Based on the study of embodied carbon transfers in trade of 185 countries in 2016, the following suggestions are provided.

Firstly, developed regions are accustomed to importing carbon-intensive low-value-added products from developing countries and backward low-income regions. Large amounts of carbon emissions have been transferred to developing countries through trade, and these regions have become global “pollution shelters”. Effective global carbon policy requires global participation, which can only be widely accepted by countries in the world based on equitable carbon emission responsibilities for both developed and developing countries. Carbon emission responsibilities should be distributed among developed and developing country/regions to truly promote global emissions reductions. In conclusion, no matter what kind of principle in terms of global carbon emission division is chosen, countries should firmly follow the path of being resource-saving, environment-friendly, low-carbon, and ecological civilized. So that carbon emissions can be minimized, the process of climate change can be slowed down and sustainable economic, social, environmental, resource, and ecological five-in-one sustainable development can be effectively achieved.

Secondly, countries around the world should actively adjust the export structure based on their own embodied carbon transfers in trade. More importantly, it is necessary to actively develop their own industries and emission reduction technologies, such as Carbon Capture, Utilization and Storage (CCUS), strive to upgrade their position in the global industrial chain, and gradually shift from exporting primary and carbon-intensive products to exporting high-value products. Many developing countries, which are unlikely to abandon the advantages of traditional resources coal and oil in the short term, can try to develop technologies (such as CCUS) to convert fossil energy into “zero carbon” energy. It is supposed for them to increase investment in zero carbon technology and enhance competitiveness in the global carbon neutral era. In addition, the government should also strengthen the top-level design to increase carbon taxes for enterprises and force high-carbon enterprises to take the initiative to reduce emissions by increasing the cost of carbon emissions.

Thirdly, importing and exporting countries can detect the transfers of embodied carbon by establishing indicators for embodied carbon transfers. On a global scale, the United States, France, and South Korea have participated in the development of carbon labeling systems. It is necessary for other countries/regions to learn from developed countries to adjust the structure of their exports and imports to reduce their pressures of carbon emission reductions, curb exports of carbon-intensive commodities, and promote exports of high-tech products. With the construction of more effective cooperation and coordination mechanisms, the cooperation among developed and developing countries should be further established. Due to the slow progress in carbon reduction by their own efforts, developing and underdeveloped countries/regions need more aid and support in global trade. Moreover, in order to construct an effective institutional tool to regulate embodied carbon transfers in international trade, the construction and improvement of the trading market in terms of global carbon emission is supposed to be accelerated and the measures of market mechanism and government regulation need to be combined.

## Figures and Tables

**Figure 1 ijerph-19-14605-f001:**
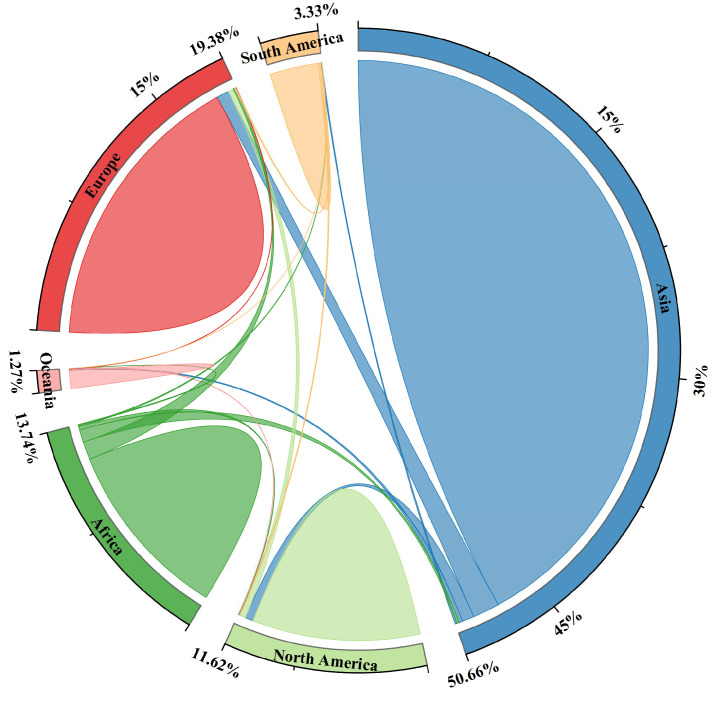
Embodied carbon transfers from trade among six continents (blue represents the carbon outflow of Asia, green represents the carbon outflow of Africa, and the same for other colors).

**Figure 2 ijerph-19-14605-f002:**
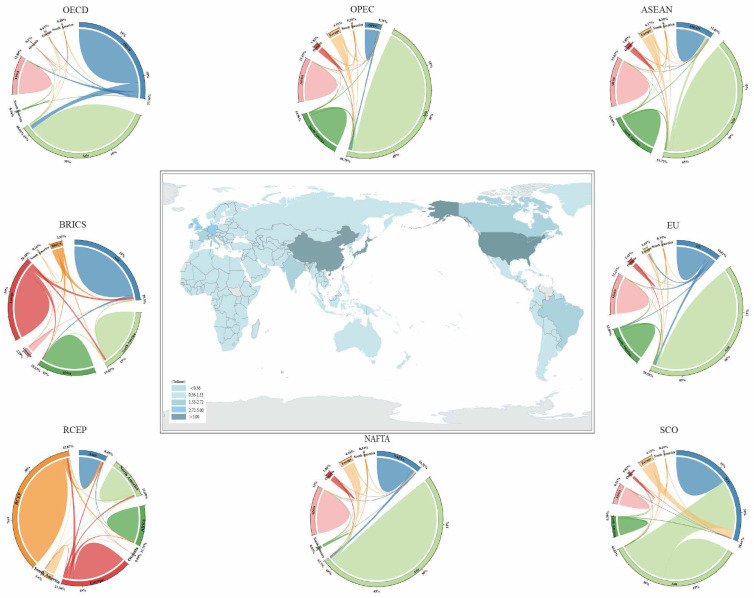
Maps of global GDP (in current dollars) and embodied carbon transfers among eight major economic cooperation organizations.

**Figure 3 ijerph-19-14605-f003:**
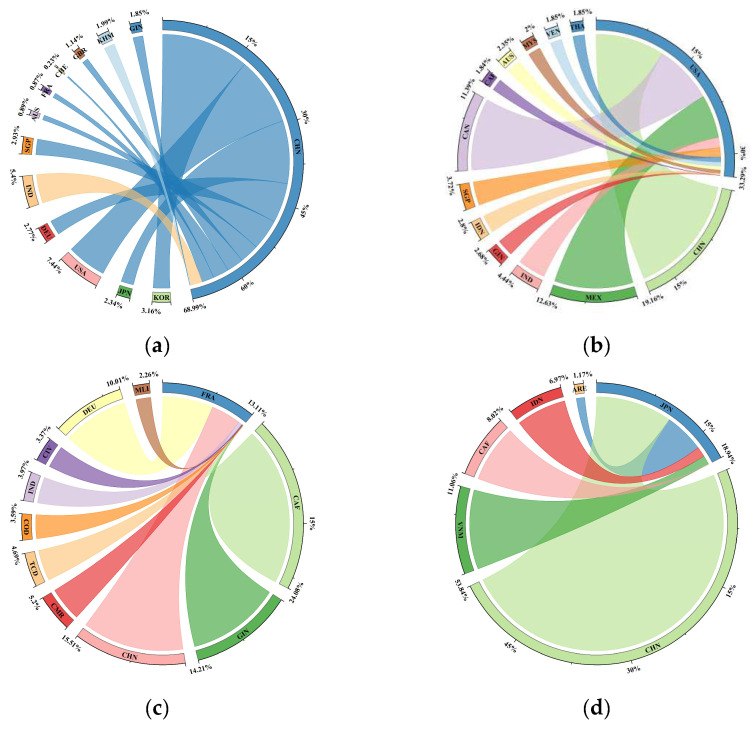
Embodied carbon transfers of major economies. ((**a**): China; (**b**): United States; (**c**): France; (**d**): Japan; (**e**): German; (**f**): Russia).

**Figure 4 ijerph-19-14605-f004:**
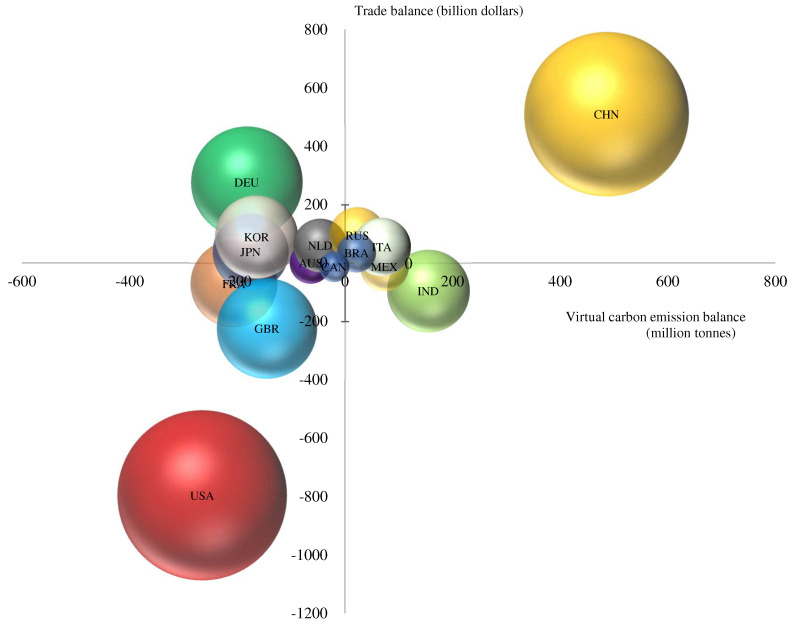
Trade and carbon trade balances for major economies.

**Table 1 ijerph-19-14605-t001:** MRIO table. The MRIO table is divided into R regions; each region has N sectors. Z represents intermediate inputs, Y represents final demand, X represents final output, V represents added value, and Q represents direct carbon emissions.

						Country 1		Country R		
Input	Output	Country 1		Country R		Final Demand		Final Demand		Gross Output
		Sector 1	Sector N	Sector 1	Sector N	Sector 1	Sector M	Sector 1	Sector M	
Country 1	sector 1	$z_{i}^{ef}$				$y_{i}^{ef}$				$x_{i}^{e}$
	...									
	sector N									
...										
Country R	sector 1									
	...									
	sector N									
Country 1	Value added	$v_{i}^{ef}$								
...	...									
Country R	Value added									
Environmental account	CO_2_	$q_{i}^{e}$								

**Table 2 ijerph-19-14605-t002:** Classification of regions.

Abbreviation	Countries or Regions in Eora Global MRIO
Asia	Yemen, Vietnam, Uzbekistan, UAE, Turkmenistan, Turkey, Thailand, Syria, Sri Lanka, Singapore, Saudi Arabia, South Korea, Qatar, Philippines, Pakistan, Oman, Gaza Strip, Nepal, Myanmar, Mongolia, Maldives, Malaysia, Lebanon, Laos, Kyrgyzstan, Kuwait, Kazakhstan, Jordan, Japan, Israel, Iraq, Iran, Indonesia, Georgia, North Korea, Cyprus, Cambodia, Brunei, Bhutan, Bangladesh, Bahrain, Azerbaijan, Armenia, Afghanistan, China, India.
North America	USA, Trinidad and Tobago, Panama, Nicaragua, Mexico, Jamaica, Hungary, Honduras, Haiti, Guatemala, Greenland, El Salvador, Dominican Republic, Cuba, Costa Rica, Cayman Islands, Canada, British Virgin Islands, Belize, Barbados, Bahamas, Antigua, Albania.
Africa	Zimbabwe, Zambia, Tanzania, Uganda, Tunisia, Togo, Tajikistan, Sudan, South Sudan, Somalia, Sierra Leone, Seychelles, Senegal, Sao Tome and Principe, Rwanda, Nigeria, Namibia, Niger, Mozambique, Morocco, Mauritius, Mrt, Mauritania, Malawi, Madagascar, Libya, Liberia, Lesotho, Kenya, Guinea, Ghana, Gambia, Gabon, Ethiopia, Egypt, Eritrea, Djibouti, DR Congo, Cote d’Ivoire, Congo, Chad, Central African Republic, Cameroon, Cape Verde, Burundi, Burkina Faso, Botswana, Benin, Angola, Algeria, South Africa.
Oceania	Vanuatu, Samoa, Papua New Guinea, New Zealand, New Caledonia, French Polynesia, Fiji, Australia.
Europe	UK, Ukraine, TFYR Macedonia, Switzerland, Sweden, Swaziland, Spain, Slovenia, Slovakia, Serbia, San Marino, Romania, Portugal, Poland, Norway, Netherlands, Montenegro, Malta, Monaco, Luxembourg, Lithuania, Latvia, Liechtenstein, Italy, Ireland, Iceland, Greece, Germany, France, Finland, Estonia, Denmark, Czech Republic, Croatia, Bulgaria, Bosnia and Herzegovina, Bolivia, Bermuda, Belgium, Belarus, Austria, Andorra, Russia.
South America	Venezuela, Uruguay, Peru, Suriname, Paraguay, Netherlands Antilles, Guyana, Ecuador, Colombia, Chile, Aruba, Argentina, Brazil.
HAD	Austria, Belgium, Bulgaria, Cyprus, Croatia, Czech Republic, Denmark, Estonia, Finland, France, Germany, Greece, Hungary, Ireland, Italy, Latvia, Lithuania, Luxembourg, Malta, Netherlands, Poland, Portugal, Romania, Slovakia, Slovenia, Spain, Sweden, the United Kingdom.
ASEAN	Brunei, Cambodia, Indonesia, Laos, Malaysia, Myanmar, Philippines, Singapore, Thailand, Vietnam.
BRICS	China, Russia, Brazil, South Africa, India.
RCEP	China, Japan, South Korea, New Zealand, Australia, Brunei, Cambodia, Indonesia, Laos, Malaysia, Myanmar, Philippines, Singapore, Thailand, Vietnam.
OIL	USA, Mexico, Canada.
OECD	USA, UK, France, German, Canada, Italy, Ireland, Netherlands, Belgium, Luxembourg, Austria, Switzerland, Norway, Iceland, Denmark, Sweden, Spain, Portugal, Greece, Turkey, Japan, Finland, Australia, New Zealand, Mexico, Czech Republic, Hungary, Poland, South Korea, Slovakia, Chile, Slovenia, Estonia, Israel, Latvia, Lithuania, Colombia, Costa Rica.
OPEC	Algeria, Iran, Iraq, Kuwait, Libya, Nigeria, Saudi Arabia, UAE, Venezuela, Angola, Gabon, Guinea, Congo.
SCO	China, Russia, Kazakhstan, Kyrgyzstan, Tajikistan, Uzbekistan, Pakistan, India.

## Data Availability

The data used to support the findings of this study are available from the corresponding authors upon request.

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
