# Peer review of "Trade and Embodied CO2 Emissions: Analysis from a Global Input–Output Perspective"

_ijerph, 2022, doi:10.3390/ijerph192114605_

Round 1
Reviewer 1 Report
Dear authors,
After reviewing the manuscript I have found it very interesting and useful for a better understanding of the international trade flows and their implications on CO2 emissions. This is in itself more than enough reason to consider it an interesting and necessary contribution.
However, I find some aspects and lacks that should be improved in order to consider its potential publication.
In my opinion, the aim of the paper is not sufficiently clear and the research developed is too descriptive and lacks depth in the analysis and interpretation of the explanatory factors. A research paper should attempt to provide more than a description or comparison of a phenomenon. It should raise questions and hypotheses, trying to explain the factors behind the observed facts.
This general observation (deficiency) is accompanied by the following specific issues:
- The topic and international approach are very interesting, considering different geographical levels (continents, cooperation organizations, countries), but the analysis made is basically descriptive. A deeper analysis, considering factors explaining international disparities in embodied CO2 emissions is missed.
- An inter-temporal analysis (evolution of trade and embodied CO2 emissions) is missing in the research (The research focuses only in 2016). In this sense, the analysis of trends is often more relevant than the analysis of the differences existing at a given time, precisely because they reflect changes in strategies or factors.
- The Section 1 (Introduction) should be improved by including references of other works that analysed this topic at global or regional level. A deeper literature review is necessary.
- In the Section 2 (Methods), the description of methods seems to be too basic. Equations 1-5 are a basic approach to the I-O methodology (essential and general equations). The methods description should focus in the development of equation 5 and next, which are the part directly linked to the research developed in the paper.
- The Figures are fine in general, but they are not well placed in the text. Figures should be included after the text that refers to them and not before. In addition, Fig 2 is too compressed and the size of each graphic should be increased.
- The text should include specific references of works analysing each continent or country (e.g. regarding European Union: Rodil-Marzábal, Óscar, and Hugo Campos-Romero. 2021. "The Intra-EU Value Chain: An Approach to Its Economic Dimension and Environmental Impact" Economies 9, no. 2: 54. https://doi.org/10.3390/economies9020054).
- It should be avoided paragraphs too long; consider to divide them into 2 or more paragraphs. This problem is repeated throughout the manuscript.
- Some ideas in the text should be clarified. For example, the phrase "China is in the middle and lower reaches of the low value chain, while Germany is the upstream high value chain of the global trade market" should be further explained and supported by evidence or work.
Reviewer 2 Report
The article makes an analysis on the trade embodied carbon transfers of bilateral trade between 185 countries/regions around the world by using a multi-regional input output model MRIO. Although the article shows that “there is still a lack of analysis of the transfer characteristics of the embodied carbon in global trade.”, there’s no relevant analysis, specific angle or tools for the characteristics mentioned, which makes the results limited to superficial comparisons between countries, and lacking key analyses for characteristic such as impact effects, causes, trends and paths, etc.
The review of embodied carbon field and analysis method is scarce, also the most frequently cited literature is not referred (e.g. Sato, 2014), not only for the embodied carbon, but also for the commonly used models, the pros and cons of MRIO, etc. Also, for the description of the method needs to be more hierarchical and the equation needs to be quoted in the text. For the result, more detailed information such as the framework of global MRIO, the digital value of embodied carbon of different sectors of countries, etc. are also needed.
Round 2
Reviewer 1 Report
Dear Authors,
Thank you very much for your efforts. I have verified that all my comments and suggestions on the first version of the manuscript are addressed in this revised version. Congratulations, I find the new version more clear and readable. Best wishes
Author Response
Thank you again for your valuable advice.